# Acceptability and perceived barriers to adoption of the core outcome set for maternal and neonatal health research and surveillance during emerging and ongoing epidemic threats (MNH-EPI-COS): An online survey

Karen Klein[1]*, Juan Pedro Alonso[2,3], Mabel Berrueta[1], Olufemi T. Oladapo[4], Mercedes Bonet[4], María Belizán[2‡], Verónica Pingray[1‡]

1 Department of Maternal and Child Health Research, Institute for Clinical Effectiveness and Health Policy, Buenos Aires, Argentina, 2 Unit of Qualitative Health Research, Institute for Clinical Effectiveness and Health Policy, Buenos Aires, Argentina, 3 Consejo Nacional de Investigaciones Científicas y Técnicas, Buenos Aires, Argentina, 4 UNDP-UNFPA-UNICEF-WHO-World Bank Special Programme of Research, Development and Research Training in Human Reproduction (HRP), Department of Sexual and Reproductive Health and Research, World Health Organization, Geneva, Switzerland

☯ KK and JPA contributed equally and share first co-authorship.
‡ VP and MBel contributed equally and share senior co-authorship.
* kklein@iecs.org.ar

## Abstract

The Maternal and Newborn Health Core Outcome Set during Epidemics (MNH-EPI-COS) is a standardized set of outcomes developed to harmonize outcome selection in maternal and neonatal health research conducted during outbreaks and epidemics. It was developed through a four-stage modified Delphi process involving a large group of international stakeholders who assessed outcomes relevance through online surveys, followed by consensus meetings with a subgroup of stakeholders to finalize the COS. The objective of this study is to evaluate the acceptability of the full MNH-EPI-COS among key stakeholders who participated in the first two round of the Delphi process, to identify anticipated barriers to its adoption, and to assess agreement on the inclusion of individual outcomes, their definitions, and the perceived feasibility of data collection. An online consultation was conducted using an electronic semi-structured survey targeting senior clinical and public health experts and civil society representatives who had contributed to earlier phases of MNH-EPI-COS development but did not participate in the final consensus meetings. Of the 118 invited stakeholders, 100 completed the survey. The majority (95%) agreed that the MNH-EPI-COS captures the most important outcomes, is likely acceptable to key stakeholders (94%), and facilitates timely evidence generation (92%). Additionally, 75% expressed intent to use it. Over 80%

**Data availability statement:** All relevant data are within the paper and its Supporting information files.

**Funding:** This work was supported by the Bill & Melinda Gates Foundation (grant INV-041181)(https://www.gatesfoundation.org/) and the UNDP/UNFPA/UNICEF/WHO/World Bank Special Programme of Research, Development and Research Training in Human Reproduction (HRP) (https://www.who.int/teams/sexual-and-reproductive-health-and-research-(srh)/human-reproduction-programme), a cosponsored programme executed by the World Health Organization (HQHRP2422779). MBo and OTO contributed to funding acquisition.

**Competing interests:** The authors have declared that no competing interests exist.

of participants agreed with the individual outcomes and their definitions, except for "skin-to-skin contact" and "breastfeeding," which were acceptable to 67% and 74%, respectively. Concerns were raised about the feasibility of measuring specific outcomes across diverse settings due to the substantial effort and resources required. Key barriers to adoption include knowledge, skills, and understanding gaps and the lack of practical resources. The MNH-EPI-COS, including its outcomes and definitions, was highly acceptable to the larger group of stakeholders involved in the early stages of its development. However, feasibility concerns remain. Successful implementation will require effective dissemination, targeted training, data collection resources, and real-world evaluation.

## Introduction

In recent decades, numerous infectious disease outbreaks with epidemic and pandemic potential have significantly impacted pregnant and recently pregnant women, fetuses, and neonates. The variability in how health outcomes are defined, measured, and reported across studies during these outbreaks delays timely evidence generation and informed decision-making [1–3]. Core Outcome Sets (COS) have been developed to promote standardized measurement and harmonized reporting in all studies for specific topics [4,5]. The COMET (Core Outcome Measures in Effectiveness Trials) Initiative provides guidance on a multi-step process for developing COS, which includes engaging key stakeholders to elicit views and prioritize important outcomes through iterative online surveys and a final in-person consensus meeting [5].

During COS development, only a subset of participants from earlier online surveys typically participates in the final in-person meeting. This contrasts with other traditional consensus methods that typically maintain a constant sample size throughout multiple process rounds. The smaller group involved in the COS final in-person meeting may not adequately represent broader stakeholder perspectives, thus introducing potential representations and selection biases. Additionally, the dynamics of in-person, non-anonymous meetings can influence the consensus process. The vocal presence of specific individuals may disproportionately impact the voting behaviour of others, while spoken and non-verbal communication can skew the results, aligning them more closely with dominant opinions rather than true collective agreement [6]. In addition, when participants with diverse backgrounds are involved, the voices of civil society representatives are often marginalized [7], being overshadowed by professionals who exert a more decisive influence during discussions [8].

Although COMET guides this process, published evidence regarding the validity of the methods and significance of the consensus meeting remains scarce, which may also pose barriers to the subsequent adoption of the COS [6,9]. Beune et al. [6] added an online consultation round to two COS projects to validate decisions made in in-person meetings, proposing this step to enhance face validity and improve COS

adoption. Laureij et al. validated the Pregnancy and Childbirth (PCB) outcome set by assessing its applicability among end-users before implementation [9].

Another critical component for adopting the COS is overcoming the barriers to its use. These barriers include limited awareness of COS, challenges in finding, selecting, and using COS, difficulties and cost to measure outcomes, and trialists' preference to use their prioritized outcomes [10]. While the literature has explored barriers to COS adoption in trials, it remains unclear whether observational and surveillance studies will encounter different challenges, particularly in the context of epidemics [11–16].

The MNH-EPI-COS was developed to evaluate maternal and neonatal health during emerging and ongoing epidemic threats across epidemiological studies, clinical studies assessing the safety and effectiveness of preventive and therapeutic interventions, and post- authorization safety surveillance [7]. The development process, adhering to COMET guidelines, employed the Delphi methodology through online surveys involving a broad group of stakeholders, followed by consensus meetings with a smaller subset of participants who finalized the COS and established agreed-upon definitions for each outcome. In this study, we aim to evaluate the acceptability of the final MNH-EPI-COS among participants who took part in the initial Delphi surveys but did not attend the consensus meetings. We assessed the overall acceptability of the final full MNH-EPI-COS, identified anticipated barriers to its adoption, and assessed agreement with the inclusion of individual outcomes, their definitions, and the perceived feasibility of data collection.

## Materials and methods

### Study overview

We conducted a cross-sectional study by administering an international online survey among participants involved in the earlier phases of the MNH-EPI-COS development process.

### Summary of MNH-EPI-COS development process

The development process of the MNH-EPI-COS is described in detail elsewhere [7]. Briefly, the MNH-EPI-COS was developed through an iterative four-stage modified Delphi process with 140 international stakeholders representing diverse professional backgrounds, genders, and world regions. Participants were senior professionals with proven experience in research, programmatic roles, or policy development related to epidemics, alongside representatives from organizations advocating for maternal and neonatal well-being.

A list of outcomes identified through a systematic review of observational and experimental epidemic-related studies was evaluated in two rounds of online surveys, in which stakeholders rated their relevance, followed by two consensus meetings to finalize the COS with a subgroup of 24 stakeholders selected to represent diverse regions and specialties.

During these meetings, survey results were thoroughly discussed, and critical decisions were made to finalize the MNH-EPI-COS, including selecting main and complementary outcomes and agreement on definitions for all outcomes. Complementary outcomes were introduced to address specific needs based on the outbreak's study type, setting, or nature, acknowledging that some outcomes might be impractical to measure in resource-limited settings or irrelevant to specific pathogens.

The final MNH-EPI-COS [7] includes seven primary maternal outcomes, 11 primary neonatal outcomes, seven complementary maternal outcomes, and four complementary neonatal outcomes. The complete MNH-EPI-COS is presented in Box 1.

S2 File provides a summarized overview of the methodology used throughout the development of the MNH-EPI-COS and this consultation.

**Box 1. Final MNH-EPI-COS.**

| Main outcomes (for all studies) | |
|---|---|
| **Maternal** | **Neonatal** |
| • Pregnancy outcome<br> • Live birth<br> • Stillbirth<br> • Miscarriage<br> • Induced abortion<br>• Maternal death<br>• Maternal suspected symptomatic infection (related outbreak disease)<br>• Maternal confirmed infection (related outbreak disease)<br>• Maternal severe disease (related outbreak disease)<br>• Preterm delivery<br> • Spontaneous<br> • Iatrogenic<br>• Mode of birth | • Neonatal death<br> • Early<br> • Late<br>• Neonatal suspected symptomatic infection (related outbreak disease)<br>• Neonatal confirmed infection (related outbreak disease)<br>• Neonatal severe disease (related outbreak disease)<br>• Vertical transmission<br>• Low birth weight<br>• Prematurity<br>• Any congenital disorder<br>• Neonatal respiratory support<br>• Skin-to-skin contact within the first hour after birth<br>• Breastfeeding within the first hour after birth |
| Complementary outcomes (for specific study type, setting and infectious disease) | |
| • Antepartum haemorrhage<br>• Postpartum haemorrhage<br>• Hypertensive disorders of pregnancy<br>• Maternal sepsis<br>• Maternal admission to ICU/special units<br>• Maternal respiratory support<br>• Maternal depression and anxiety | • Neonatal admission to NICU/special units<br>• Neonatal respiratory failure<br>• Birth Asphyxia<br>• Neonatal sepsis |

## Acceptability online survey participants

We invited all stakeholders (N = 118) who completed the first two online Delphi surveys during the initial development of the COS but did not attend the final consensus meetings to complete an additional online survey to assess the acceptability of the MNH-EPI-COS.

## Acceptability online survey development and administration

A semi-structured questionnaire was designed, including closed and open-ended questions. The questionnaire was piloted among seven participants who closely matched the target participant profile. The survey items were developed based on a comprehensive literature review of frameworks and validated questionnaires for measuring acceptability [17–19], and studies on barriers to COS adoption [9,11–16,20]. The survey focused on three aspects: a) the overall acceptability of the MNH-EPI-COS, b) anticipated barriers to its adoption in observational research studies and surveillance in the context of epidemics, and c) individual outcome assessment.

   The online survey (via Survey Monkey) was self-administered and anonymous (S1 File). The overall acceptability of the full MNH-EPI-COS was evaluated using five items on a 5-point Likert scale. These items assessed agreement with the core outcomes, perceived acceptability, potential to facilitate timely evidence generation for decision-making, effort required for use, and intention to use. Participants were invited to provide additional comments on the overall acceptability of the MNH-EPI-COS in free-text fields. The survey also explored anticipated barriers to adopting the MNH-EPI-COS for observational research and surveillance in the context of epidemics. It included predefined options based on barriers identified in the literature and structured according to the COM-B model [21,22]. Finally, for individual core outcomes, participants were asked to assess the acceptability of including it in the COS as a main or complementary outcome, the acceptability of the provided definition, and the feasibility of data collection using a 5-point Likert scale (strongly disagree, disagree, neither agree nor disagree, agree, and strongly agree). Participants also had the option to select "unable to assess." Participants were encouraged to explain their reasoning in an open-ended field if they disagreed or strongly

disagreed with any item. Respondents were able to review and change their answers; unique visitor was determined by IP address, and the survey never displayed a second time once the user had filled it. Demographic data, including participants' main roles, specialties, and genders, were collected. Before the survey, participants received a summary of MNH-EPI-COS development methods and results, including the final list of main and complementary maternal and neonatal outcomes, with their agreed-upon definitions.

An invitation to participate in the survey was sent on August 31–118 stakeholders. Reminders were sent to participants with partial or no responses over 6 weeks (August 31 to October 7, 2024). Participants provided electronic informed consent before completing the survey.

### Ethics statements

This study involved the same stakeholders who participated in earlier consultation rounds during the development of the MNH-EPI-COS [7,23], which was granted an exception by the WHO Ethics Review Committee, as the project posed no risk of harm to participants. All participants provided electronic written informed consent before completing the survey (for further details, refer to S1 File, Page 2).

### Analysis

Quantitative data were analysed using descriptive statistics, with the results presented as percentages and absolute numbers. For overall acceptability, we report the percentages for all options on a Likert scale. For individual outcomes, the acceptance rate was calculated as the percentage of participants who selected "strongly agree" or "agree" out of all participants who assessed that outcome. Acceptability was defined as agreement by at least 80% of the participants following the agreement threshold used in developing the MNH-EPI-COS.

Open-ended responses were analyzed using a qualitative thematic analysis. Comments for each open-ended survey question were coded and analyzed separately to identify question-specific themes. Two independent coders systematically reviewed the responses and developed inductive coding frameworks tailored to each question. Discrepancies in coding were resolved through discussion and consensus. The emerging themes were then reviewed and refined in collaboration with the broader research team to ensure interpretive rigor, address potential biases, and enhance the validity of the findings. This approach allowed us to capture nuanced insights into participants' perspectives on the overall acceptability of the MNH EPI COS, as well as their specific concerns regarding the inclusion, definition, or feasibility of individual outcomes.

## Results

### Study population

Out of the 118 invited stakeholders, 100 responded to the survey (85% response rate), with balanced representation across professional backgrounds, roles, specialties, genders, and geographical regions (Table 1 and S2 File).

### Overall acceptability of the MNH-EPI-COS

Ninety-five percent of participants agreed that the MNH-EPI-COS captures the most important outcomes, 94% agreed that it is likely acceptable to key stakeholders, and 92% agreed that it facilitates timely evidence generation during outbreaks. When asked about their intention to use the COS, 75% agreed or strongly agreed.

Regarding the effort required to use the COS, only 21% indicated it would require little to no effort, while 73% anticipated a moderate to high effort (Table 2). In open-ended comments, some participants observed that the perceived effort needed to implement the MNH-EPI-COS was significantly impacted by the number of included outcomes and practical challenges related to data collection feasibility.

**Table 1. Characteristics of participants.**

| Characteristics | % of participants (N = 100) |
|---|---|
| **Main role** | |
| Researcher | 46% |
| Healthcare provider | 35% |
| Women/community representative | 10% |
| Policy maker | 3% |
| Funder | 3% |
| Health service manager | 2% |
| Program Manager | 1% |
| **Main specialties&** | |
| Maternal health | 55% |
| Neonatal and paediatric health | 25% |
| Epidemiology and public health | 24% |
| Infectious disease | 12% |
| Patient, women, and community advocacy | 10% |
| Pharmacy/Laboratory | 5% |
| Critical care | 4% |
| Psychiatry/psychology/social work | 3% |
| Anthropology | 1% |
| Physiology | 1% |
| **WHO Region** | |
| Americas | 29% |
| Europe | 26% |
| Africa | 15% |
| Western Pacific | 15% |
| Eastern Mediterranean | 11% |
| South-East Asia | 4% |
| **Gender** | |
| Female | 67% |
| Male | 33% |
| **Age** | |
| <30 | 1% |
| 30-39 | 9% |
| 40-49 | 34% |
| 50-59 | 34% |
| >=60 | 22% |

&Participants could report more than one specialty.

## Anticipated barriers to MNH-EPI-COS adoption

The anticipated barriers to adopting the MNH-EPI-COS in observational studies and surveillance are presented in Table 3 and organized according to the COM-B model. Most anticipated barriers fall under the Capability and Physical Opportunity domains. Within Capability—which includes the knowledge, skills, and understanding required for effective implementation—the most frequently cited barriers were 'Lack of awareness of the COS' (64% of participants) and 'Poor knowledge and understanding of COS' (44% of participants).

**Table 2. Overall acceptability of the MNH-EPI-COS.**

| | % of participants (N = 100) |
|---|---|
| **To what extent do you agree that the MNH-EPI-COS captures the most important outcomes?** | |
| Strongly agree | 52% |
| Agree | 43% |
| Neither agree nor disagree | 3% |
| Disagree | 1% |
| Strongly disagree | 0% |
| Don´t know | 1% |
| **The MNH-EPI-COS is likely to be acceptable to key stakeholders.** | |
| Strongly agree | 40% |
| Agree | 54% |
| Neither agree nor disagree | 3% |
| Disagree | 1% |
| Strongly disagree | 0% |
| Don´t know | 2% |
| **To what extent do you agree that the MNH-EPI-COS will facilitate timely evidence generation for decision-making during outbreaks and epidemics?** | |
| Strongly agree | 44% |
| Agree | 48% |
| Neither agree nor disagree | 4% |
| Disagree | 1% |
| Strongly disagree | 0% |
| Don´t know | 3% |
| **I intend to use the MNH-EPI-COS for maternal and neonatal health research and surveillance in the context of epidemics.** | |
| Strongly agree | 34% |
| Agree | 41% |
| Neither agree nor disagree | 13% |
| Disagree | 0% |
| Strongly disagree | 1% |
| Don´t know | 11% |
| **How much effort will it take to use the MHN-EPI-COS?** | |
| No effort at all | 2% |
| A little effort | 19% |
| Moderate effort | 48% |
| A lot of effort | 19% |
| Huge effort | 6% |
| Don´t know | 6% |

The Physical Opportunity domain highlights the role of external resources and operational requirements required for MNH-EPI-COS adoption. The 'Resource requirement associated with measuring the COS' (64%) was considered a significant barrier. A common concern was a lack of guidance and validated tools to collect data and measure outcomes (47%).

Participants elaborated on anticipated barriers in open-ended fields, particularly regarding the challenges faced in low- and middle-income countries (LMICs), highlighting issues such as limited resources, inadequate infrastructure (e.g., health information systems), and insufficient training. Participants also emphasized the importance of working on the

**Table 3. Anticipated barriers to MNH-EPI-COS adoption in observational research studies and surveillance in the context of epidemics.**

| Barriers | % of participants prioritizing each barrier (N = 100) |
|---|---|
| **Capability** | |
| Lack of awareness of the COS | 64% |
| Poor knowledge and understanding of COS | 44% |
| Lack of skills to apply the COS | 37% |
| Difficulties/challenges choosing between multiple COS | 24% |
| **Opportunity: Physical** | |
| Resource requirements associated with measuring the COS | 64% |
| Lack of guidance and validated tools to collect data and measure outcomes | 47% |
| Increased burden on researchers and patients due to the data collection required to report COS | 39% |
| Too many COS outcomes limit the reporting of all outcomes | 36% |
| COS not accessible in locally understandable languages | 30% |
| Lack of harmonization of some outcome definitions | 1% |
| **Opportunity: Social** | |
| Lack of external incentives (e.g., recommendations by funders or regulatory agencies) | 32% |
| Lack of engagement of decision-makers | 1% |
| **Motivation** | |
| Lack of perceived usefulness in generating timely evidence for decision-making | 32% |
| Researcher preference to choose their own outcomes | 27% |
| COS could be seen as restrictive and limiting the range of outcomes | 11% |
| Concerns about the quality of the COS development and methods for keeping it updated | 10% |
| A limited number of patient-centred outcomes | 10% |

Note: Participants could select one or more anticipated barriers.

dissemination of the MNH-EPI-COS, securing support from key governmental stakeholders, and providing targeted training in data collection and COS implementation for healthcare providers.

### Individual outcome assessment

Table 4 shows the acceptability of (a) the inclusion of each outcome, categorized as main or complementary; (b) each outcome definition; and (c) the perceived feasibility of data collection for each outcome included in the MNH-EPI-COS.

**Main maternal and neonatal outcomes.** The inclusion of individual outcomes as main outcomes was acceptable to more than 80% of participants, except for two neonatal outcomes: ´skin-to-skin contact during the first hour after birth´, which was acceptable to 67% of participants, and ´Breastfeeding within one hour of birth', which was acceptable to 74% of participants. For those who did not agree with the inclusion of these outcomes, only 9% of participants explicitly disagreed with the inclusion of 'Skin-to-skin contact', and 6% disagreed with 'breastfeeding'. In comparison, 23% and 20% remained neutral (neither agreeing nor disagreeing). All representatives of women and civil society (n = 10) agreed to include both outcomes.

**Table 4.** Acceptability for the inclusion of outcomes and definitions and feasibility of data collection for each outcome.

| | Acceptability to include this outcome as a main or complementary outcome | | Acceptability with the proposed definition | | Perceived feasibility of collecting this outcome | |
|---|---|---|---|---|---|---|
| **Maternal main outcomes** | | | | | | |
| *Mortality/vital status* | | | | | | |
| Pregnancy outcome | 97/100 | 97% | 90/97 | 93% | 87/99 | 88% |
| Maternal death | 98/100 | 98% | 95/100 | 95% | 92/100 | 92% |
| *Maternal infection (related outbreak disease)* | | | | | | |
| Maternal suspected symptomatic infection (related outbreak disease) | 89/100 | 89% | 88/100 | 88% | 78/98 | 80% |
| Maternal confirmed infection (related outbreak disease) | 93/990 | 94% | 90/98 | 92% | 75/97 | 77% |
| Maternal severe disease (related outbreak disease) | 95/100 | 95% | 90/99 | 91% | 77/96 | 80% |
| *Labour and delivery characteristics* | | | | | | |
| Spontaneous/iatrogenic preterm birth | 95/100 | 95% | 94/100 | 94% | 80/99 | 81% |
| Mode of birth | 87/100 | 87% | 89/100 | 89% | 88/99 | 89% |
| **Neonatal main outcomes** | | | | | | |
| *Mortality/vital status* | | | | | | |
| Neonatal death | 99/100 | 99% | 92/98 | 94% | 93/98 | 95% |
| *Neonatal infection* | | | | | | |
| Neonatal symptomatic infection (related outbreak disease) | 93/100 | 93% | 91/98 | 93% | 78/97 | 80% |
| Neonatal confirmed infection (related outbreak disease) | 95/99 | 96% | 92/98 | 94% | 83/97 | 86% |
| Neonatal severe/critical disease (related outbreak disease) | 93/100 | 93% | 86/98 | 88% | 80/97 | 82% |
| Vertical transmission | 89/99 | 90% | 88/99 | 89% | 61/96 | 64% |
| *Morbidity* | | | | | | |
| Low birthweight | 97/100 | 97% | 95/100 | 95% | 94/99 | 95% |
| Prematurity | 97/99 | 98% | 91/96 | 93% | 85/99 | 86% |
| Any congenital anomaly | 90/97 | 93% | 89/96 | 93% | 69/96 | 72% |
| *Delivery of care* | | | | | | |
| Neonatal respiratory support | 87/98 | 89% | 87/97 | 90% | 78/97 | 80% |
| Skin-to-skin contact during the first hour after birth | 66/98 | 67% | 79/97 | 81% | 61/98 | 62% |
| Breastfeeding within one hour of birth | 73/99 | 74% | 84/97 | 85% | 61/99 | 62% |
| **Maternal complementary outcomes** | | | | | | |
| *Morbidity* | | | | | | |
| Antepartum haemorrhage | 84/98 | 86% | 82/96 | 85% | 73/97 | 75% |
| Postpartum haemorrhage | 92/99 | 93% | 86/96 | 90% | 79/98 | 81% |
| Hypertensive disorders of pregnancy | 89/100 | 89% | 78/96 | 81% | 73/98 | 74% |
| Maternal sepsis | 98/100 | 98% | 92/96 | 96% | 87/98 | 89% |
| *Delivery of care* | | | | | | |
| Maternal admission to intensive care unit/ special unit | 96/98 | 98% | 92/96 | 96% | 87/97 | 90% |
| Maternal respiratory support | 97 | 90% | 93/96 | 97% | 81/97 | 84% |
| *Maternal Functioning* | | | | | | |
| Maternal symptoms of depression and anxiety | 80/100 | 80% | 79/96 | 82% | 53/99 | 54% |
| **Neonatal complementary outcomes** | | | | | | |
| **Morbidity** | | | | | | |
| Birth asphyxia | 87/99 | 88% | 78/92 | 85% | 79/97 | 81% |
| Neonatal sepsis | 92/98 | 94% | 83/95 | 87% | 74/96 | 77% |
| Neonatal respiratory failure | 90/98 | 92% | 91/95 | 96% | 77/96 | 80% |

*(Continued)*

**Table 4.** (Continued)

| | Acceptability to include this outcome as a main or complementary outcome | | Acceptability with the proposed definition | | Perceived feasibility of collecting this outcome | |
|---|---|---|---|---|---|---|
| **Delivery of care** | | | | | | |
| Neonatal admission to the intensive care unit/other special units | 91/99 | 92% | 94/98 | 96% | 87/99 | 88% |

Note: Denominators varied if some participants reported: "Unable to assess." Values are highlighted in green when acceptability is => 80% and red when acceptability is below 80%.

Participants who disagreed with including these outcomes noted that they are often absent from routine datasets or find it difficult to collect valid measures, especially during outbreaks. Additionally, they pointed out that practices vary widely across settings, leading to potential inconsistencies and inaccuracies in reporting, and that these outcomes may be of lower priority than other main neonatal outcomes. The participants also highlighted overlaps between the two outcomes, as breastfeeding typically involves skin-to-skin contact.

About the acceptability of the definitions, all main outcomes demonstrated acceptance levels above 80%, with most approaching 90%.

Most of these outcomes were deemed feasible for collection with approximately 80% agreement. However, one maternal outcome and four neonatal outcomes fell below this threshold: 'Maternal confirmed infection' (77%); 'Any congenital anomaly' (72%); 'Vertical transmission' (64%); 'Skin-to-skin contact during the first hour after birth' (62%); and 'Breastfeeding within one hour of birth' (62%).

The reasons for lower feasibility were concerns owing to resource constraints and limitations in existing datasets, with additional potential challenges during outbreaks when data collection practices may be disrupted. 'Maternal confirmed infection' was identified as a challenge due to limited access to tests, laboratory supplies, and trained personnel —issues particularly emphasized in LMICs— and the potential lack of confirmatory testing during early-stage outbreaks. For 'Any congenital anomalies', participants disagreed due to the absence of routine monitoring systems in some countries. Vertical transmission challenges include confirming cases, identifying transmission routes, and detecting transmission at time points beyond in-utero exposure. 'Skin-to-skin contact' and 'Breastfeeding' are often not part of routine datasets, and collecting data on these outcomes is particularly challenging during outbreaks.

**Complementary maternal and neonatal outcomes.** The acceptability of individual outcome inclusion and their definitions exceeded 80% for all complementary outcomes. On the other hand, less than 80% of participants agreed with the feasibility of collecting three maternal and one neonatal outcome. More specifically, 'Antepartum haemorrhage' and 'Hypertensive disorders of pregnancy' had 75% agreement on feasibility, while 'Maternal symptoms of depression and anxiety' showed 50% agreement, and 'Neonatal sepsis' was 77%.

The reasons for the disagreement on feasibility centred on challenges related to data reliability, consistency across settings, and resource limitations. For 'Antepartum haemorrhage', concerns were raised about the potential for over-reporting and inconsistent quantification across settings. Insufficient blood pressure monitoring due to resource constraints, out-of-facility births, or disruptions to prenatal care posed challenges for 'Hypertensive disorders of pregnancy'. Regarding 'Maternal symptoms of depression and anxiety', disagreements were linked to the infrequent inclusion of mental health data in routine datasets and variability in the capacity to assess mental health outcomes. For 'Neonatal sepsis', reasons for disagreement included limited diagnostic skills and inadequate laboratory capacity in some contexts.

## Discussion

### Main findings and interpretation

This study aimed to evaluate the acceptability and perceived feasibility of the MNH-EPI-COS by the broader group of stakeholders involved in the early stages (online surveys) of the development of the COS. These stakeholders found the MNH-EPI-COS highly acceptable and agreed that the set includes the most important outcomes. Additionally, most participants deemed the individual outcomes and their proposed definitions acceptable—however, some anticipated challenges related to the feasibility of data collection. The feasibility of measuring specific outcomes across diverse contexts remains a significant concern, with participants highlighting the substantial efforts required for successful implementation.

The high acceptability of the MNH-EPI-COS, including its outcomes and definitions, aligns with the findings of Beune et al., [6] who evaluated the face validity of decisions made during consensus meetings by a larger group of participants through online surveys of two COS in the field of maternal and foetal health. Their results demonstrated high concordance between the larger Delphi panel and the smaller consensus group, affirming the representativeness of the meeting's decisions. Another study [9] used a mixed-methods design to validate a pregnancy and childbirth outcome set by assessing its relevance across all end users before implementation. This study supported the applicability of the outcome set, with most survey participants agreeing that the set contained the most important outcomes.

Despite the overall acceptability of the MNH-EPI-COS, two neonatal outcomes— '*Skin-to-skin contact*' and '*Breastfeeding within one hour of birth*'—garnered lower levels of agreement for inclusion as main outcomes due to perceived redundancy, lower relevance compared to clinical outcomes, and data collection feasibility concerns. Similar resistance emerged during the early rounds of the COS development process, where these outcomes initially received lower scores but were ultimately retained due to strong advocacy by civil society representatives and the use of specific mechanisms—such as independent consultations—designed to elevate their voices without the influence of professional stakeholders [7,23] These challenges echo the findings of Laureij et al., [9] who reported variability in the perceived relevance of patient-reported outcome measures in a COS on pregnancy and childbirth care. They also resonate with broader discussions in the literature regarding the challenges of avoiding the dilution of civil society voices, such as patients and caretakers, in the COS development. Significant barriers include power and representativeness imbalances between lay and expert stakeholders, physicians overlooking patient-relevant topics or outcomes, challenges engaging and sustaining patient participation, and difficulties identifying effective methods to capture patients' perspectives [24–27]. As Chevance et al. [28] argue, selecting outcomes for inclusion in a COS is not merely a scientific exercise but also an ethical and political one guided by societal values and public health needs. Recognizing and addressing these dimensions is essential to ensure that a COS reflects diverse perspectives and serves as a tool for equitable healthcare improvement. Meaningful civil society participation—achieved via independent consultations, inclusive recruitment from advocacy networks, and community engagement frameworks preventing professional dominance—is critical to creating COSs that embody diverse priorities and foster equitable, person-centered healthcare.

While most outcomes were deemed feasible to collect, some participants perceived some as challenging. Feasibility concerns were primarily attributed to resource constraints and limitations in existing datasets. Such feasibility challenges have been well-documented in the literature. For example, during infectious outbreaks, the lack of well-equipped laboratories and trained personnel has been identified as a significant barrier to accurately measuring 'Maternal confirmed infection.'[27] Some outcomes are also underreported in routine data collection, as is the case with congenital anomalies, which are missing from the records of more than half of LMICs due to outdated reporting formats and delays in documenting findings from first- and second-trimester ultrasonography [29,30].

Finally, the anticipated barriers to adopting the MNI-EPI-COS in observational studies and surveillance align with those identified in the literature on COS adoption in trials [11–16]. Most barriers expected are related to the knowledge, skills, and understanding required for effective implementation, as well as the practical resources and logistical demands

involved. These results suggest that limited awareness of the COS, insufficient knowledge of its purpose and benefits, and a lack of training and resources for its implementation could hinder its broader adoption.

To ensure the widespread adoption and impact of the MNH-EPI-COS, it is essential to implement a robust dissemination and communication strategy complemented by targeted education and training interventions. Developing a comprehensive data collection toolkit with accessible and standardized tools, such as case report forms with clear definitions, is critical for consistent data collection and outcome measurements. Addressing resource limitations through strategic funding and partnerships will also be crucial for the success of these efforts. These tools and resources should be developed to effectively tackle feasibility challenges associated with measuring outcomes across diverse settings, thereby enhancing usability and practical implementation. Additionally, as this study focused on assessing the acceptability of the COS and anticipated barriers among participants involved in its development, further evaluation in real-world settings is necessary. Such implementation research is essential to assess the feasibility, usability, and accuracy of outcome reporting, address context-specific challenges, and build confidence among a broader range of end-users to adopt the COS.Updates to the COS are essential to maintain its relevance, requiring periodic review and revision based on emerging evidence, evolving clinical practices, and stakeholder feedback. To address this, regular review by the MNH-EPI-COS steering committee, informed by pilot testing and field implementation, is key to ensuring its long-term relevance and usability. Together, these steps will facilitate MNH-EPI-COS implementation and maximize its impact on evidence generation and informed decision-making.

## Strengths and limitations

A key strength of our study was its high response rate, which enhances the robustness of the process and ensures confidence that the MNH-EPI-COS reflects a broad, consensus-based perspective. Additionally, the sample included stakeholders with diverse disciplinary backgrounds and geographic regions, enriching the breadth of perspectives from individuals working in varied settings.

However, the study has some limitations. First, while we made efforts to include participants from different regions and professional backgrounds, certain populations - such as non-English speakers and stakeholders from rural or marginalized communities -, were underrepresented or excluded from the survey. We recognize the ethical implications of this limitation and the importance of ensuring that future assessments are more inclusive and accessible. Second, the extraction of items to assess acceptability and the response options for evaluating potential barriers to COS adoption was not based on a systematic literature review. Nonetheless, we employed established theoretical frameworks to ensure a structured and conceptually grounded approach to survey development. In addition, including open-ended questions enabled us to capture novel insights, such as unanticipated barriers, feasibility concerns, and the rationale behind disagreements on outcome inclusion. Third, the survey focused on hypothetical scenarios, including anticipated barriers to adopting the COS, the effort required for its implementation, and feasibility assessments. This forward-looking approach can be particularly challenging, as the MNI-EPI-COS has not yet been implemented, and the actual difficulties encountered during implementation may differ from those anticipated. Identifying potential barriers and implementation challenges offers valuable guidance for refining strategies and enhancing COS adoption; these insights must, however, be further explored and approached through piloting the COS in real-world settings across diverse contexts to uncover unforeseen obstacles and ensure broader applicability.

## Conclusions

The overall MNH-EPI-COS, including its outcomes and definitions defined in consensus meetings, is deemed acceptable to a large group of key stakeholders. Although data collection appears feasible, there are challenges in measuring some outcomes and anticipated barriers to COS adoption. Addressing these challenges is crucial to ensure the successful adoption of the MNH-EPI-COS. The development of a practical implementation toolkit is strongly recommended to support

the adoption and integration of the MNH-EPI-COS in diverse settings. Additionally, future efforts should evaluate the acceptability of the COS among a broader range of stakeholders to strengthen the generalizability and robustness of the findings.

## Supporting information

**S1 File. MNH_EPI _COS Acceptability Survey.**
(DOCX)

**S2 File. Additional Methods.**
(DOCX)

**S3 File. Participants of the online survey.**
(DOCX)

**S4 File. Checklist for Reporting Results of Internet E-Surveys (CHERRIES).**
(DOCX)

**S5 File. Database and Dictionary.**
(XLSX)

## Acknowledgments

The authors would like to express their gratitude to the participants of the consultation survey (S3 File) for their invaluable time and input, as well as to the participants of the pilot survey: Jackeline Alger, Maria Fernanda Escobar Vidarte, Sofia Castiglioni, Sandra Formia, Alessandra L. Marcone, Rangel Mirna Montenegro, and Vanesa Ortega. Special thanks to Gabriela Radice for her comprehensive literature review on COS adoption barriers and Anna Portnoy's insightful contributions in editing the manuscript. The authors also acknowledge the support of the WHO for this project.

## Author contributions

**Conceptualization:** Mabel Berrueta, Mercedes Bonet, Verónica Pingray.

**Data curation:** Karen Klein, Juan Pedro Alonso, María Belizán, Verónica Pingray.

**Formal analysis:** Karen Klein, Juan Pedro Alonso, María Belizán, Verónica Pingray.

**Funding acquisition:** Olufemi T. Oladapo, Mercedes Bonet.

**Investigation:** Karen Klein, Juan Pedro Alonso, María Belizán, Verónica Pingray.

**Methodology:** María Belizán, Verónica Pingray.

**Project administration:** Karen Klein, Mercedes Bonet, Verónica Pingray.

**Resources:** Olufemi T. Oladapo, Mercedes Bonet.

**Supervision:** Karen Klein, Olufemi T. Oladapo, Mercedes Bonet, Verónica Pingray.

**Validation:** Olufemi T. Oladapo, Mercedes Bonet.

**Visualization:** Karen Klein.

**Writing – original draft:** Karen Klein, Juan Pedro Alonso, Verónica Pingray.

**Writing – review & editing:** Karen Klein, Juan Pedro Alonso, Mabel Berrueta, Olufemi T. Oladapo, Mercedes Bonet, María Belizán, Verónica Pingray.

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
