## [Decision Letter · Decision Letter 0]

10 Jun 2025

PGPH-D-25-00952

Acceptability and perceived barriers to adoption of the core outcome set for maternal and neonatal health research and surveillance during emerging and ongoing epidemic threats (MNH-EPI-COS). An online survey

Dear Dr. Klein,

Thank you for submitting your manuscript to PLOS Global Public Health. After careful consideration, we feel that it has merit but does not fully meet PLOS Global Public Health’s publication criteria as it currently stands. Therefore, we invite you to submit a revised version of the manuscript that addresses the points raised during the review process.

We look forward to receiving your revised manuscript.

Kind regards,

Muhammad Imran Nisar, MBBS, MSc

Academic Editor

Journal Requirements:

1. We have amended your Competing Interest statement to comply with journal style. We kindly ask that you double check the statement and let us know if anything is incorrect.

2. In the online submission form, you indicated that Data are available upon reasonable request. 

3. Uploaded as supplementary information.

2. We do not publish any copyright or trademark symbols that usually accompany proprietary names, eg (R), (C), or TM  (e.g. next to drug or reagent names). Please remove all instances of trademark/copyright symbols throughout the text, including ™ on page 8.

Additional Editor Comments (if provided):

Reviewers' comments:

Reviewer's Responses to Questions

**Comments to the Author**

1. Does this manuscript meet PLOS Global Public Health’s publication criteria?

Reviewer #1: Partly

Reviewer #2: Yes

2. Has the statistical analysis been performed appropriately and rigorously?

Reviewer #1: I don't know

Reviewer #2: Yes

3. Have the authors made all data underlying the findings in their manuscript fully available (please refer to the Data Availability Statement at the start of the manuscript PDF file)?

Reviewer #1: Yes

Reviewer #2: No

4. Is the manuscript presented in an intelligible fashion and written in standard English?

Reviewer #1: Yes

Reviewer #2: Yes

Reviewer #1: Dear authors,

The study provides valuable insights into COS acceptability but requires stronger methodological rigor, real-world validation, and actionable implementation strategies. Addressing these critiques would enhance its impact and applicability across diverse settings.

Accordingly, please respond to the following:

1.The study participants were limited to stakeholders involved in the earlier phases of the MNH-EPI-COS development, potentially introducing selection bias. The absence of new or independent stakeholders may skew the results toward pre-existing biases. Including a broader range of stakeholders, such as frontline healthcare workers and policymakers not previously involved in the COS development, would eventually validate the findings more robustly.

2.The survey items were not based on a systematic literature review, which could lead to gaps in addressing all relevant barriers and acceptability factors. Please conduct a systematic review to identify all potential barriers and facilitators of COS adoption, ensuring the survey comprehensively covers these aspects.

3.The feasibility of data collection was assessed hypothetically, as the COS had not yet been implemented. This may not accurately reflect real-world challenges. It was better to pilot the COS in a few diverse settings to gather empirical data on feasibility and barriers before finalizing the survey results. I suggest marking this note as a limitation of your work.

4.The qualitative analysis of open-ended responses lacks detail on methodology (e.g., coding framework, inter-rater reliability), raising concerns about the rigor and reproducibility of the findings. Please provide a detailed description of the qualitative analysis process, including how themes were identified and validated, to enhance transparency.

5.Many outcomes (e.g., "maternal confirmed infection," "congenital anomalies") were deemed less feasible in low-resource settings due to limited infrastructure and training. This is another limitation to be documented at the end of your work.

6.The paper acknowledges civil society voices were marginalized in consensus meetings but does not explore how this might affect COS adoption or equity. I encourage you to include a deeper analysis of power dynamics and strategies to amplify underrepresented voices (e.g., community engagement frameworks).

7.The study does not address how the COS will be updated or maintained over time to remain relevant. Based on that, recommend a governance structure (e.g., steering committee) to review and update the COS periodically based on new evidence and stakeholder feedback.

8.Ethical implications of excluding certain populations (e.g., non-English speakers, rural communities) from the survey are not discussed. I invite you to address ethical gaps by ensuring future surveys are multilingual and inclusive of marginalized groups to enhance equity.

Reviewer #2: Manuscript id: PONE-D-25-00952

General Comments:

The study titled Acceptability and perceived barriers to adoption of the core outcome set for maternal and neonatal health research and surveillance during emerging and ongoing epidemic threats (MNH-EPI-COS). An online survey

Understanding the acceptability and barriers to adopting the core outcome set for maternal and neonatal health during epidemics (MNH-EPI-COS) is critical for effective global health responses. This research identifies perceived challenges that could prevent adoption of core outcome set of standardized data collection (for example during crises like COVID-19), potentially hampering evidence-based decision-making and cross-study comparison. This paper is connected to the published article of the same research team: “A core outcome set for maternal and neonatal health research and surveillance of emerging and ongoing epidemic threats (MNH-EPI-COS): a modified Delphi-based international consensus; Pingray, Verónica et al.; eClinicalMedicine, Volume 80, 103025”

I am recommending publishing this manuscript with minor changes.

Abstract: It is understandable that MNH-EPI-COS is prominent acronym, however, author can elaborate it first time in the abstract.

Analysis:

Line 159: Authors can more elaborate on the process of qualitative analysis

References:

Line 354: Authors could provide DOI or PMID number of all references.

Conclusion

Line 339: Authors can re-suggest in the conclusion as “Recommendation for development of toolkit is important for adoption and implementation”.

Other comments:

-No information on the data availability and not as per the requirement of PLOS data availability policy.

**Do you want your identity to be public for this peer review?** For information about this choice, including consent withdrawal, please see our Privacy Policy

Reviewer #1: **Yes: ** Hayder Al-Momen

Reviewer #2: No

---

## [Decision Letter · Decision Letter 1]

31 Oct 2025

Acceptability and perceived barriers to adoption of the core outcome set for maternal and neonatal health research and surveillance during emerging and ongoing epidemic threats (MNH-EPI-COS). An online survey

PGPH-D-25-00952R1

Dear Dr Klein,

We are pleased to inform you that your manuscript 'Acceptability and perceived barriers to adoption of the core outcome set for maternal and neonatal health research and surveillance during emerging and ongoing epidemic threats (MNH-EPI-COS). An online survey' has been provisionally accepted for publication in PLOS Global Public Health.

Best regards,

Julia Robinson

Executive Editor

Reviewer Comments (if any, and for reference):

Reviewer's Responses to Questions

**Comments to the Author**

Reviewer #1: All comments have been addressed

publication criteria?

Reviewer #1: Yes

3. Has the statistical analysis been performed appropriately and rigorously?

Reviewer #1: Yes

4. Have the authors made all data underlying the findings in their manuscript fully available (please refer to the Data Availability Statement at the start of the manuscript PDF file)?

Reviewer #1: Yes

5. Is the manuscript presented in an intelligible fashion and written in standard English?

Reviewer #1: Yes

Reviewer #1: Thank you for your professional responses.

**Do you want your identity to be public for this peer review?** For information about this choice, including consent withdrawal, please see our Privacy Policy

Reviewer #1: **Yes: ** Hayder Al-Momen
